# Methodological Approaches to DNA Authentication of Foods, Wines and Raw Materials for Their Production

**DOI:** 10.3390/foods10030595

**Published:** 2021-03-11

**Authors:** Aram G. Galstyan, Vladislav K. Semipyatniy, Irina Yu. Mikhailova, Khamid Kh. Gilmanov, Alana V. Bigaeva, Ramil R. Vafin

**Affiliations:** 1All-Russian Research Institute of the Dairy Industry, Lusinovskaya str. 35, 115093 Moscow, Russia; a_galstyan@vnimi.org (A.G.G.); vniipbivp@fncps.ru (I.Y.M.); kh_gilmanov@vnimi.org (K.K.G.); a_bigaeva@vnimi.org (A.V.B.); r_vafin@vnimi.org (R.R.V.); 2All-Russian Scientific Research Institute of Brewing, Beverage and Wine Industry, Rossolimo str. 7, 119021 Moscow, Russia

**Keywords:** *Vitis vinifera* L, wine, sample preparation, authentication, DNA extraction and amplification, *UFGT*, SNP

## Abstract

DNA authentication of wines is a process of verifying their authenticity by genetic identification of the main plant component. The sample preparation of experimental and commercial wines was carried out by precipitation of wine debris by centrifugation with preliminary exposure with precipitators and co-precipitators, including developed macro- and micro-volume methods applicable to white or red wines, using polyvinylpyrrolidone as a co-precipitator. Addition of 2-mercaptoethanol and proteinase K to the lysing solution made it possible to adapt the technology for DNA extraction from the precipitated wine debris. The additionally tested technique of DNA extraction from wine debris by dimethyl sulfoxide (DMSO) lysis had fewer stages and, consequently, a lower risk of contamination. The results of further testing of one of the designed primer pairs (UFGT-F1 and UFGT-R1) in conjunction with the tested methods of wine material sample preparation and nucleic acid extraction, showed the advantage in the given set of oligonucleotides over previously used ones in terms of sensitivity, specificity and reproducibility. The developing strategy for genetic identification of grape varieties and DNA authentication of wines produced from them based on direct sequencing of polymerase chain reaction (PCR) products is implemented by interpreting the detected polymorphic positions of variable *Vitis vinifera* L. *UFGT* gene locus with distribution and split into 13 *UFGT* gene-associated groups.

## 1. Introduction

The search for objective criteria for assessing the authenticity and the place of origin of food products with the possibility of traceability of the entire life cycle “from field to counter” is a strategically important task that can only be achieved by multidisciplinary science-intensive approaches [1].

The development of genomic, post-genomic and bioinformatic technologies increase the potential of molecular genetic approaches to DNA authentication of products and raw material for their production, unfolding perspectives of pilot design/elaboration in the quality management system with monitoring of counterfeit products, including wine [2].

DNA authentication of wines—a technological process of verification of their authenticity by genetic identification of the main plant component—technical varieties of grapes, molecular genetic analysis of residual nucleic acids of *Vitis vinifera* L., extracted also from precipitated wine debris formed at the stage of sample preparation of biomaterial under research [3].

Various methods of sample preparation of wines based on centrifugation of debris precipitation with an additional use of precipitants and co-precipitants are known in the art.

F. Savazzini and L. Martinelli (2006) [4] performed sample preparation of commercial wines by precipitation of wine debris by centrifugation with preliminary exposure of the wine in a mixture with NaCl or isopropanol or sodium acetate in a freezer (−20 °C) for 1–2 weeks. From the listed precipitators, L. Pereira et al. (2011) [5] opted for isopropanol and a 2-week exposure at a similar temperature, and V. Catalano et al. (2016) [6] reproduced this proven method of sample preparation in their research.

J. Drábek et al. (2008) [7] performed sample preparation of commercial wines by centrifugal precipitation of wine debris with preliminary short-term exposure at room temperature of a mixture of wine with ethylenediaminetetraacetic acid (EDTA) and linear polyacrylamide, as well as supplementary addition of MgCl_2_ or NaCl in combination with isopropanol.

J. Bigliazzi et al. (2012) [8] used a combination of two precipitators—isopropanol and sodium acetate for aging commercial and experimental wines with night exposure of the resulting mixtures at −20 °C and further centrifugation, whereas R. Vignani et al. (2019) [2] aged the mixture in similar proportions of precipitators at −80 °C for 3 days.

Isci B. et al. (2014) [9] precipitated relatively large volumes (from 100 to 750 mL) of stum and wine by high-speed centrifugation, or smaller volumes of the sample (45 mL) mixed with NaCl after a week’s exposure in the freezer. The applied principle of centrifugal precipitation without precipitators had a certain similarity with the previously proposed approaches described in the research works of R. Siret et al. (2000) [10] and E. Garcia-Beneytez et al. (2002) [11].

The article by C. Agrimonti and N. Marmiroli (2018) [12] provides a summary of the methods of DNA extraction from precipitated wine debris with references to a number of scientific articles that also reflect the relevant sample preparation procedures.

The most common approaches to the extraction of nucleic acids from precipitated wine debris are methods based on a modification of the well-known cetyltrimethylammonium bromide (CTAB) method, which is based on cell lysis with a buffer with an active compound of CTAB, deproteinization with chloroform, and DNA precipitation with isopropanol [3].

Lysis of resuspended sediment is usually carried out by a multicomponent buffer system, which includes such reagents as EDTA—ethylenediaminetetraacetic acid, Tris-HCl—Tris(hydroxymethyl)aminomethane hydrochloride, NaCl—sodium chloride, CTAB –cetyltrimethylammonium bromide, PVP—polyvinylpyrrolidone, 2-mercaptoethanol) [4,8,10], including proteinase K [5,6].

In the basic list of CTAB-modified methods for extracting nucleic acids of *Vitis vinifera* L. from wines can be attributed to conditionally named approaches Siret [10], Savazzini and Martinelli [4], Pereira [5] and Bigliazzi [8] in original and optimized productions [6], which have similar and distinctive features.

Along with the CTAB method, commercial nucleic acid extraction kits were also used, such as the “QIAprep Spin Miniprep Kit” (Qiagen) [8,10,13], “DNeasy Plant Mini Kit” (Qiagen) [4,11,14,15], “NucleoSpin Food” (Macherey-Nagel) [16], “QuickExtract Seed DNA Extraction Solution” (Epicentre Biotechnologies) [17].

In comparative testing of many commercial kits for DNA extraction V. Catalano et al. (2016) [6] indicated a certain effectiveness of some of them: “Dneasy Plant mini kit” (Qiagen), “High Pure polymerase chain reaction (PCR) template preparation kit” (Roche) and “Power Soil DNA isolation kit” (MO-BIO) subject to their minor modifications associated with the addition of α-amylase and/or proteinase K, to improve the performance of testing systems.

Using nuclear microsatellite loci (nSSR) [18,19,20] and chloroplast (cpSSR) [21,22,23] DNA which were intended for genetic identification of grapes variety is one of the approaches to DNA authentication of wines [1,2,3,4,5,6,8,9,10,11,12,13,17,24,25,26], a sequential procedure which is carried out with separately taken SSR-marker primer set, and not multiplex PCR.

From the variety of highly polymorphic nSSR markers, identifiable results of a number of markers are noted: VrZAG79 [5,6], VVS2, VVMD27 [4,8], and VVMD25 [8], which are appropriate for DNA authentication of wine with interpretation of data from fragmentary analysis of the capillary sequencer.

Less polymorphic cpSSR markers of chloroplast DNA, despite their weak discrimination ability, have greater abundance on the cell, greater resistance to exonucleases and less susceptibility to degradation due to their presence in double-membrane organelles, which is especially important when working with residual nucleic acid of *Vitis vinifera* extracted from wines [6].

Together with SSR markers, SNP markers have a high identification potential that are suitable for genetic identification of *Vitis vinifera* L. varieties and DNA authentication of wines produced from them by interpreting the results of sequencing and/or analysis of high-resolution melting curves (HRM analysis) on PCR platforms with fluorescence hybridization detection [27,28,29,30,31].

The purpose of the research is to create scientific and methodological approaches to DNA authentication of wine products and the main raw material for its production.

In accordance with the purpose of the research work, the following tasks were set:to test well-known and developed methods for sample preparation and extraction of nucleic acids of the *Vitis vinifera* L. from raw grapes and wines produced from them;to develop a strategy for their DNA authentication by interpreting the detected polymorphic positions of variable *Vitis vinifera* L. *UFGT* gene by direct sequencing of the PCR product.

## 2. Materials and Methods

The research was conducted in the laboratory of the Interdisciplinary scientific-technical center of food products quality control of the All-Russian Research Institute of Brewing, Non-Alcoholic and Wine Industry.

Objects of research: technical (wine) grape varieties, experimental young wines, commercial white and red monosort dry wines of protected geographical indications and protected names of places of origin.

The work involves molecular genetic research methods aimed at sample preparation and extraction of nucleic acids from research objects, amplification and sequencing of generated PCR products, as well as bioinformatic and mathematical analysis, including the development of a specialized database in the context of DNA authentication of wine products and raw material for its production.

Sample preparation of technical grape varieties was carried out by extracting 50–100 mg of grape pulp adjacent to the exocarp, with further placement of the selected sample of plant tissue in a 1.5 mL Eppendorf-type test tube.

Sample preparation of experimental and commercial wines was carried out by precipitation of wine debris by centrifugation with preliminary exposure with precipitators and co-precipitators. Isopropanol (I) was used as precipitators in combination with sodium acetate (II), in combination with co-precipitators—linear polyacrylamide (III) and polyvinylpyrrolidone (IV), as part of sample preparation in a 15 mL Falcon-type test tube. Sample preparation in a 1.5 mL Eppendorf-type test tube was performed in a mixture of red wine with isopropanol and polyvinylpyrrolidone (V).

A visual diagram of the components of the tested sample preparation methods (I-IV) is shown in Table 1.

Nucleic acids were extracted from the grape pulp using a commercial “DNA-Sorb-S-M kit” (Central research Institute of Epidemiology, Russia), and from the precipitated wine debris—a modification of this kit with adding of 4 µL of 2-mercaptoethanol and 100 µL of proteinase K (10 mg/mL) to the lysing solution. In addition, the technique of DNA extraction from wine debris by dimethyl sulfoxide (DMSO) lysis has been tested.

The stages of the tested methods for extracting nucleic acids from wine debris are shown in Table 2.

PCR with extracted DNA samples was performed using a commercial reagent “Encyclo Plus PCR kit” (Evrogen JSC, Moscow, Russia), which includes the necessary components of the reaction mixture (sterile water, buffer system, a mixture of deoxynucleotide (dNTP) and Encyclo polymerases), with the exception of oligonucleotide primers.

The corresponding protocols for preparing reaction mixtures for PCR with DNA samples extracted from grapes and wine debris are presented in Table 3.

Table 4 shows the list of selected primers and thermal cycling modes used for setting up PCR for genetic identification of grape varieties and DNA authentication of wines produced from them.

Electrophoretic detection of amplified PCR products was performed in 2.5% agarose gel on a tris-acetate-EDTA (TAE) buffer stained with ethidium bromide, followed by DNA visualization in a UV transilluminator (Vilber Lourmat, Collégien, France) and recording the result on a digital camera.

Amplicons of the analyzed *UFGT* gene locus generated by the UFGT-F1 and UFGT-R1 oligonucleotides (and further used as sequence primers) were sequenced on the “ABI PRISM 3100 genetic analyzer” (Applied Biosystems, Foster City, CA, USA), followed by their alignment in Basic Local Alignment Search Tool (BLAST) with the corresponding partial nucleotide sequences of *Vitis vinifera* L.

## 3. Results

At the first stage of molecular genetic studies, two technical grape varieties of Chardonnay and Cabernet Sauvignon were sampled by extracting grape pulp, as well as sample preparation of experimental young wines produced from them by precipitation of wine debris by centrifugation with preliminary exposure with isopropanol (method I, Table 1) during one day in the freezer.

The sample prepared material was then subjected to the procedure of extraction of nucleic acids by a commercial “DNA-sorb-S-M kit” (without modification), followed by the use of extracted DNA samples in PCR.

The electrophoretic picture of the first results of PCR amplification of the *Vitis vinifera* L. *UFGT* and *F3H* gene loci initiated with well-known primer sets is shown in Figure 1.

The results of preliminary studies indicated that the selected conditions for sample preparation, DNA extraction, and PCR were applicable for amplification of the analyzed gene locus directly to the raw material, but not to the wine material produced from it.

Thus, PCR of DNA samples extracted from the pulp of Chardonnay and Cabernet Sauvignon grape varieties produced a specific amplification of 705 bp initiated with primers Vv1-Fwd and Vv1-Rev [30]; 375 bp with background non-specificity (F3H_H1fwd and F3H_H1rev [31]); 119 bp (Vv3-Fwd and Vv3-Rev [30]); 532 bp (F3H_H2fwd and F3H_H2rev [31]), respectively. At the same time, specific amplification products were not developed in PCR of DNA samples extracted from the precipitated wine debris (Figure 1).

Further experiments with testing other methods of sample preparation of wine material (methods II–V, Table 1) and extraction of nucleic acids with a “DNA-Sorb-S-M kit” in modification with addition of 2-mercaptoethanol and proteinase K to the lysing solution, did not lead to satisfactory PCR results with the sets of primers described above.

Therefore, further studies were aimed at testing two designed sets of primers that initiate amplification of a specific PCR product with a length of 101 bp (UFGT-F and UFGT-R) and 99 bp (UFGT-F1 and UFGT-R1), respectively (Table 4). A schematic view of the sequences of the analyzed gene flanked by primers is shown in Figure 2.

The results of comparative testing of the designed pair of UFGT-F and UFGT-R primers with the well-known prototype (Vv3-Fwd and Vv3-Rev) are shown in Figure 3.

The obtained results of oligonucleotides testing showed a higher sensitivity of the designed set of UFGT-F and UFGT-R primers in comparison with the prototype, which is visually confirmed by the presence (Figure 3, tracks 1–4) and the absence (Figure 3, tracks 6–9) of the amplification signal in the analyzed PCR samples. At the same time, the amplification signal of a positive control DNA sample extracted from the pulp of Chardonnay grape was stronger in PCR with the UFGT-F and UFGT-R primers (Figure 3, track 5) than with the Vv3-Fwd and Vv3-Rev primers (Figure 3, track 10), which may serve as an additional justification for the sensitivity statement.

It should be noted that the combination of well-known and designed primers resulted in a positive signal of amplification of PCR samples of the analyzed DNA samples extracted from the deposited debris of white wine material rather than red ones in the Vv3-Fwd and UFGT-R pair (Figure 4).

Suspensions of linear polyacrylamide from different manufacturers used in the sample preparation method III (Table 1) did not differ significantly in the efficiency of DNA coprecipitation in a mixture with isopropanol and sodium acetate, except that Satellite Red nucleic acid co-precipitator produced by Evrogen JSC, Moscow, Russia provides easy visualization of the resulting sediment, which may be relevant when working with small volumes of precipitated white wine material and wines.

The results of further testing of another designed pair of UFGT-F1 and UFGT-R1 primers in conjunction with the tested methods of wine material sample preparation and DNA extraction showed the advantage of these oligonucleotides over previously used ones in terms of sensitivity, specificity and reproducibility.

Micro-volume rapid method for sample preparation of red wine material (method V, Table 1) and further DNA extraction by lysis of the precipitated debris with DMSO (Table 2) also led to satisfactory PCR results with UFGT-F1 and UFGT-R1 primers initiating amplification of a specific PCR product with a length of 99 bp (tracks 1–3, Figure 5).

However, another proven rapid method of sample preparation of experimental young red wine with a 10 min exposure of 1 mL of wine material mixed with 1% PVP at room temperature, followed by precipitation of wine debris by centrifugation at maximum speed of the microcentrifuge for 10 min, was ineffective (tracks 4–6, Figure 5).

It should be noted that the reproducibility of PCR with DNA samples extracted from experimental young wines was higher compared to samples of nucleic acids extracted from commercial wines (Figure 6).

As seen in Figure 6, along with the specific PCR product in one of the replications, a nonspecific shorter fragment (track 1) was also amplified, or no reaction took place at all (track 4), in case with duplicate DNA samples of the grapes Sauvignon blanc and Mencia extracted from the precipitated debris of the Visigodo and Lagar De Robla wines, respectively (Figure 6).

The negative effect of cold stabilization on the studied nucleic acid extracted from the precipitated wine debris, samples prepared from wine stored for a day in the refrigerator after uncorking the bottle, was also revealed (Figure 7).

It should be noted that the negative effect of cold stabilization on the analyzed nucleic acid, expressed in a weak amplification signal of a specific PCR product or its complete absence, was fully extended to commercial wines.

There was also poor storage in the freezer of extracted DNA samples of *Vitis vinifera* L. from the precipitated debris of commercial wines, which led to unsatisfactory results of PCR amplification of the analyzed *UFGT* gene locus. At the same time, the reproducibility of PCR with freshly extracted DNA samples that were not stored in the freezer, but were used immediately in the amplification reaction, increased.

The next stage of molecular genetic research was the sequencing of the nucleotide sequences of the *UFGT* gene locus amplified with UFGT-F1 and UFGT-R1 primers of PCR products of DNA samples extracted from the pulp of grape and precipitated wine debris.

UFGT-F1 and UFGT-R1 primers initiate the development of a specific PCR product with a length of 99–100 bp with localization in the flanked region (404–503 nt) of 5 polymorphic positions, including 1 indel (INDEL-insertion/deletion) and 4 positions of single nucleotide substitutions (SNPs—Single Nucleotide Polymorphisms), interpreted by sequencing during grape variety identification and DNA authentication of wines produced from them.

The result of alignment of reference nucleotide sequences of the *Vitis vinifera* L. *UFGT* gene locus flanked with the UFGT-F1 and UFGT-R1 primers with INDEL and SNPs is shown in Figure 8.

Good examples of detection of INDEL (424 nt) and SNPs (425, 459 and 483 nt) polymorphic positions of the *Vitis vinifera* L. *UFGT* gene locus performed by capillary Sanger sequencing method is shown in Figure 9.

Grouping and splitting of grape varieties by the profile of the analyzed polymorphic positions of the *UFGT* gene locus is one of the ways to identify them and DNA-authenticate the wines produced from them.

Bioinformatic analysis of open source data allowed us to distribute technical grape varieties with their split into 13 *UFGT* gene-associated groups (Table 5 and Table 6).

At the same time, if Table 5 provides only generalized information on a limited number of technical grape varieties of conditionally reference status with additional division by color of mature berry, then Table 6 provides already detailed information on distributed varieties in 13 *UFGT* gene-associated groups, indicating their names and countries of growth, as well as the unique identifier of the deposited nucleotide sequence (GenBank A/N).

The distribution of the studied grape varieties growing in Russia by *UFGT* gene-associated groups was as follows:

1B—Rkatsiteli (Derbent Brandy Factory), Rkatsiteli (Derbent Wine Factory, the southern part of Derbent), Riesling (Sparkling Wine Derbent Factory, the southern part of Derbent), Chardonnay (TWC), Chardonnay (Sommelier), Pervenets Magaracha (Phanagoria); 1N—Levokumsky steady (Kizlyar district, the northern part); 2B—Chinuri Gramotenko (Yalta); 2N—Malbec Gramotenko (Yalta); 4B—Muscat White (Mirny); 4N—Bastardo Gramotenko (Yalta), Yalta Vasilievsky Gramotenko (Yalta); 9N—Cabernet Sauvignon (Mirny), Cabernet Franc (Phanagoria), Cabernet Franc (Sommelier).

Sequenced PCR products of DNA samples extracted from the deposited debris of the experimental wine material had a profile of the analyzed polymorphic positions of the *UFGT* gene locus identical to the raw grape material used in their production.

However, the results of sequencing of PCR products of DNA samples extracted from the deposited debris of commercial wines did not always correspond to the variety-authentic profile, which could be caused by external and internal factors.

## 4. Discussion

Sample preparation of the biomaterial under research is the starting point, the quality of which depends on the effectiveness of the next stage—extraction of nucleic acids.

Sample preparation of technical grape varieties by extracting and transferring of a small volume of pulp to an Eppendorf-type test tube is the optimal procedure for selecting biomaterial from the raw material under study, which does not require advanced tools.

Tested methods of sample preparation of experimental and commercial wines, based on their exposure to precipitators and co-precipitators with subsequent precipitation of wine debris by centrifugation, differ in elements of components (Table 1).

If the sample preparation method I involves the use of isopropanol as the only precipitator, then in method II isopropanol is used in combination with another precipitator—sodium acetate, and in other methods this combination of precipitators works in combination with such co-precipitators as linear polyacrylamide (III) and polyvinylpyrrolidone (IV). At the same time, if the four listed methods of sample preparation (I–IV) are focused on manipulations in a 15 mL Falcon-type test tube with exposure in the freezer, then in method V all procedures are performed in a 1.5 mL Eppendorf-type test tube with further exposure of the resulting mixture at room temperature.

Vector of positive dynamics of the specified exposure time intervals for each method (Table 1) was oriented towards increasing the aging time of mainly commercial wines mixed with precipitators and co-precipitators.

It should be noted that our method of sample preparation I is similar to the method described by L. Pereira et al. (2011) [5], and the method of sample preparation II—J. Bigliazzi et al. (2012) [8], with minor changes adapted to the instrumental characteristics of the equipment in use.

Sample preparation method III, which significantly reduces the exposure time due to the use of linear polyacrylamide mixed with isopropanol and sodium acetate, had a serious drawback associated with the time-consuming procedure of resuspending the precipitated wine debris in a 15 mL Falcon-type test tube even before the transfer of the homogenate to a 1.5 mL Eppendorf-type test tube. The given method had a certain similarity to the method described by J. Drabek et al. (2008) [7] regarding the use of linear polyacrylamide and isopropanol.

Later, two methods of sample preparation using polyvinylpyrrolidone as a co-precipitator were proposed: macro-volume method IV, intended for white wines, and micro-volume method V, intended for red wines.

The minimum exposure time of the micro-volume rapid method for sample preparation of red wine material is 10 min. At the same time, small aliquots (100 µL) of samples of the wine under study mixed with isopropanol (900 µL) and PVP (10 mg) can be stored at room temperature for up to 10 days, followed by centrifugation of test tubes. This approach is an alternative to storing in the freezer the test-tube wine from an uncorked bottle [10].

Micro-volume methods for sample preparation of red stum by high-speed centrifugation of an Eppendorf-type test tube with 1.5 mL of the test sample are known in the art [4,11,12]. Whereas in this research, the possibility of sample preparation from 100 µL of red wine material using a precipitator (isopropanol) and a co-precipitator (PVP) is shown.

Nucleic acid extraction following the sample preparation tested with the commercial “DNA-sorb-S-M kit” showed the effectiveness of its use in the DNA extraction from the pulp of grape.

According to the instructions of “DNA-Sorb-S-M kit”, it is designed for DNA extraction from biological material from animals (tissue material) and food, biological additives, animal feed or plant raw material for subsequent research by polymerase chain reaction (PCR).

The principle of the extraction method is that the test sample is treated with a lysing solution with proteinase K, resulting in destruction of cell membranes, release of nucleic acids and cellular components. The dissolved nucleic acids bind to the sorbent particles, while other components of the lysed test material remain in the solution and are removed during deposition of sorbent by centrifugation and further washing of the sorbent. When buffer for elution is added to the sorbent, nucleic acids transitions from the surface of the silica into a solution, which is separated from the sorbent particles by centrifugation. As a result of the mentioned procedure, a highly purified nucleic acids preparation is obtained, free from amplification reaction inhibitors, which ensures high analytical sensitivity of the PCR study.

It should be noted that the composition of the lysis buffer reagent kit includes guanidine chloride and Triton X-100; the composition of the lysing reagent is proteinase K; the solution for washing 1 is guanidine thiocyanate and Triton X-100; the solution for washing 2 is isopropanol; the universal sorbent included in the kit is fine silicon dioxide (SiO2), and the buffer for elution B is a standard TE buffer for DNA dilution.

Modification of this kit by adding additional 2-mercaptoethanol and proteinase K to the lysing solution made it possible to adapt the technology for extraction from precipitated wine debris (Table 2), which to a certain extent overlaps with the recommendations of V. Catalano et al. (2016) [6], aimed at improving the performance of commercial testing systems.

J. Drabek et al. (2008) [7], in comparative testing of the commercial “DNAzol kit” (Invitrogen), which is a guanidine thiocyanate extraction method, showed its insignificant effectiveness over the CTAB method. Thiocyanate is present in the washing solution in the guanidine kit used by us, while another chaotropic agent (guanidine chloride) is part of the buffer for the lysing reagent.

Additionally, the proven technique of DNA extraction from wine debris by dimethyl sulfoxide (DMSO) lysis, which has fewer stages and, consequently, a lower risk of contamination, is an effective method with the prospect of its further optimization and adaptation in relation to systems for manual and automatic extraction of nucleic acids. (Table 2). The basic principle of the adapted nucleic acid extraction technique with minor modifications is borrowed from the previously published work of E. M. del Campo et al. (2010) [32].

PCR testing performed by commercial “Encyclo Plus PCR kit” (Evrogen JSC, Moscow, Russia) after the nucleic acid extraction showed the effectiveness of its use in standard protocols for preparing reaction mixtures in relation to DNA samples under research extracted from grape pulp (Table 3).

According to the technical characteristics of the used commercial kit, it is intended, including, amplification of complex mixtures of nucleic acids and low copy DNA matrices.

Protocols for preparing reaction mixtures for PCR with DNA samples extracted from the precipitated wine debris differed by a two-fold increase in a number of components (dNTP mixture, Encyclo polymerase mixture, primers) in comparison with standard protocols intended for PCR with nucleic acid samples extracted from grapes raw material (Table 3).

At the same time, if the DNA extracted from grape pulp was successfully amplified in all tested PCR protocols, including with sets of well-known primers [30,31], then the amplification of DNA extracted from the precipitated wine debris was mainly achieved only with designed primers that initiate the development of the small-sized *UFGT* gene locus (from 99 bp to 118 bp).

The inability to amplify a longer PCR product is due to significant fragmentation of the residual DNA extracted from wine debris [6]. The effectiveness of PCR drops sharply due to the fact that the DNA is fragmented to such an extent that the average size of the fragments is less than the length of the amplified gene locus.

Also, one of the reasons that did not allow us to develop a PCR product with a length of 119 bp (Figure 3, tracks 6–9) with the well-known primers Vv3-Fwd and Vv3-Rev [30] with a melting temperature difference of more than 5 °C is the relatively low DNA yield of the extraction methods used compared to Pereira method [5], which is characterized by a relatively high yield of nucleic acids extracted from the precipitated wine debris.

By combining the well-known (Vv3-Fwd) [30] and designed (UFGT-R) primers with an acceptable melting temperature difference, it became possible to obtain a positive amplification signal (118 bp) for PCR of analyzed DNA samples extracted from the precipitated white wine debris (tracks 1 and 3, Figure 4), and with two designed sets of primers UFGT-F + UFGT-R and UFGT-F1 + UFGT-R1 we were able to increase reaction sensitivity to the ability to amplify PCR products from DNA samples, including those extracted from red wine material.

At the same time, the reproducibility of PCR with DNA samples extracted from commercial wines was significantly lower compared to DNA samples extracted from experimental young wines, and the negative effect of cold stabilization on the analyzed nucleic acid often led to unsatisfactory results, which is consistent with the results of other researchers [6].

It should be taken into account that the extracted DNA of *Vitis vinifera* L. from wine has the status of a residual nucleic acid, as its concentration is significantly reduced during the multi-stage wine production process, including decanting, cleaning, filtration and other processing methods. In addition, grape DNA is degraded by DNase microbiota of wine during the fermentation process. The yield of extracted nucleic acid also decreases as the wine ages. Thus, the amount of DNA of *Vitis vinifera* L. extracted from wines with final alcoholic fermentation is significantly reduced or is practically absent, depending on the timing of testing the finished product [6,17].

In this research work the strategy of genetic identification of grape varieties and DNA authentication of wines produced from them based on direct sequencing of PCR products was implemented through the interpretation of the detected polymorphic positions of the variable *UFGT* gene locus of *Vitis vinifera* L., flanked by UFGT-F1 and UFGT-R1 primers, including those used as sequencing ones.

To “read” nucleotide sequencing from the direct primer UFGT-F1 allows to detect SNPs at positions 459 nt and 483 nt, while from the reverse primer UFGT-R1—INDEL 424 nt and SNPs 425 and 442 nt of the analyzed gene locus.

Good examples of detection of polymorphic positions by capillary Sanger sequencing indicate a possible interpretation of the data obtained in the identification key (Figure 9).

The presented result of alignment of the reference nucleotide sequences of the *Vitis vinifera* L. *UFGT* gene locus flanked with UFGT425-F1 and UFGT483-R1 primers with INDEL and SNPs (Figure 8) was generated based on the results of bioinformatic analysis of open source data, which allowed to distribute technical grape varieties with their splitting into 13 *UFGT* gene-associated groups with additional division by the color of mature berry (Table 5 and Table 6).

It should be noted that the nucleotide sequences of *UFGT* gene-associated groups #09 (Cabernet Sauvignon variety) and #10 (Souzão variety) are not deposited in GenBank NCBI are only published in a scientific article [29]. At the same time, the associated group #13 includes the nucleotide sequence of the *UFGT* gene locus of the Cabernet Sauvignon (JF522384) variety deposited in GenBank NCBI, including Cabernet Franc (JF522430).

It is worth mentioning that in this research the nucleotide sequences of the *UFGT* gene locus of the varieties Cabernet Sauvignon and Cabernet Franc growing on the territory of the Russian Federation corresponded to the associated group #09, and not #13. However, it should be noted that the detection of INDEL 424 nt is somewhat difficult, which can be misinterpreted when reading the results of Sanger sequencing on a genetic analyzer.

Sequenced PCR products of DNA samples extracted from grape raw material and experimental wines produced from them generally had an identical profile of the analyzed polymorphic positions of the *Vitis vinifera* L.*UFGT* gene locus, while the results of sequencing of commercial wines sometimes did not correspond to the variety-authentic profile due to a number of factors, one of which was the loss of one of the alleles of the heterozygous genotype gene during amplification, which ultimately led to incorrect interpretation of the data.

## 5. Conclusions

The proposed strategy of DNA authentication by interpreting the detected polymorphic positions of the variable *UFGT* gene locus of *Vitis vinifera* L. by direct sequencing of PCR product is carried out in a complex of selected conditions for sample preparation, DNA extraction, and PCR for amplification of the corresponding gene loci. Taking into account the potential and limitations of tested methods, a number of proposed techniques expanding the arsenal of methodological tools, increased the rapidity of sample preparation of wine materials and wines (with co-precipitator PVP) and extraction of residual grapevine nucleic acids (with DMSO), enhanced the sensitivity, specificity and reproducibility of PCR (with primers UFGT-F1 and UFGT-R1), as well as provided identification of the *UFGT* gene-associated groups by sequencing. In general, the analysis of well-known and developed methodological approaches to DNA authentication of wine products and raw material for its production, including those tested in this research, indicates the need for further research to improve the methods of sample preparation and nucleic acids extraction, amplification and sequencing of generated PCR products, with the involvement of other molecular genetic methods to improve the accuracy of diagnostics. At the same time, the practical significance of the conducted and planned research is aimed at implementing developments in the management control system of raw material quality and products based on developed standards for the wine industry.

## Figures and Tables

**Figure 1 foods-10-00595-f001:**
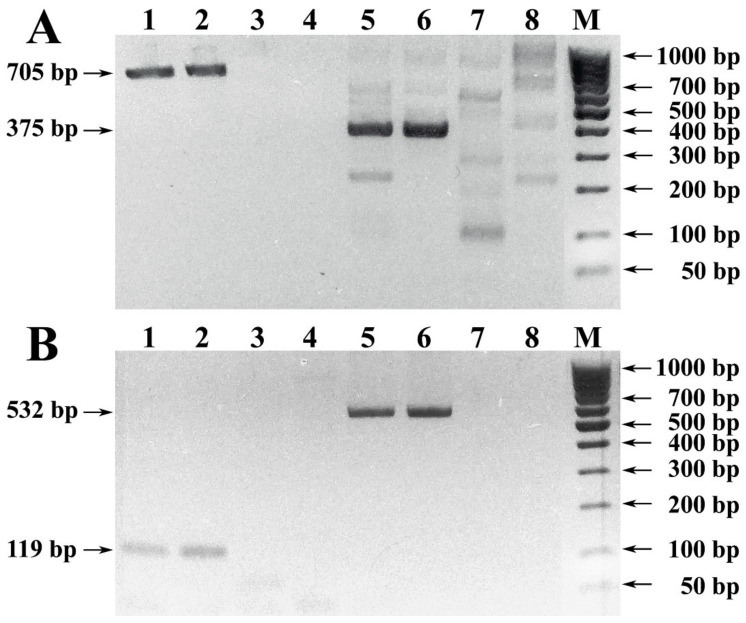
Electrophoregram of results of PCR amplification of the *Vitis vinifera* L. *UFGT* and *F3H* gene loci. Legend: (**A**1–4) Amplified with primers Vv1-Fwd and Vv1-Rev PCR samples of DNA extracted from grape pulp and precipitated wine debris: (**A**1) grape pulp of Chardonnay; (**A**2) grape pulp of Cabernet Sauvignon; (**A**3) precipitated wine debris of Chardonnay; (**A**4) precipitated wine debris of Cabernet Sauvignon. (**A**5–8) Amplified with primers F3H_H1fwd and F3H_H1rev PCR samples of DNA extracted from grape pulp and precipitated wine debris: (**A**5) grape pulp of Chardonnay; (**A**6) grape pulp of Cabernet Sauvignon; (**A**7) precipitated wine debris of Chardonnay; (**A**8) precipitated wine debris of Cabernet Sauvignon. (**B**1–4) Amplified with primers Vv3-Fwd and Vv3-Rev PCR samples of DNA extracted from grape pulp and precipitated wine debris: (**B**1) grape pulp of Chardonnay; (**B**2) grape pulp of Cabernet Sauvignon; (**B**3) precipitated wine debris of Chardonnay; (**B**4) precipitated wine debris of Cabernet Sauvignon. (**B**5–8) Amplified with primers F3H_H2fwd and F3H_H2rev PCR samples of DNA extracted from grape pulp and precipitated wine debris: (**B**5) grape pulp of Chardonnay; (**B**6) grape pulp of Cabernet Sauvignon; (**B**7) precipitated wine debris of Chardonnay; (**B**8) precipitated wine debris of Cabernet Sauvignon. (M) DNA size markers 100 bp + 50 bp (SibEnzyme).

**Figure 2 foods-10-00595-f002:**
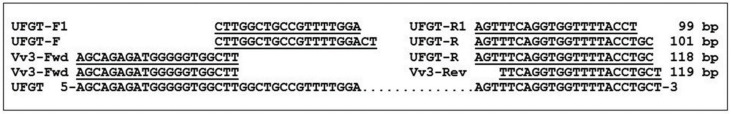
Partial nucleotide sequences of the *UFGT* gene flanked by different sets of primers.

**Figure 3 foods-10-00595-f003:**
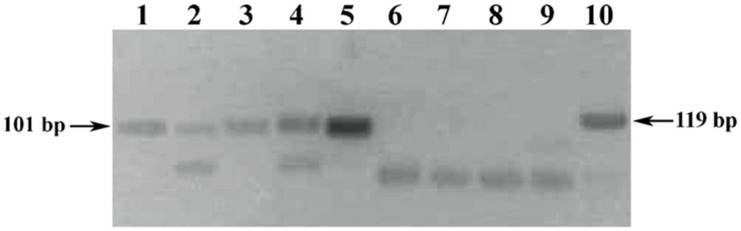
Electrophoregram of the results of PCR amplification of the *Vitis vinifera* L. *UFGT* gene locus. Legend: (1–4) PCR samples of DNA amplified with UFGT-F and UFGT-R primers, extracted from precipitated wine debris: (1–2) a 3-day exposure of wine material with isopropanol: (1) Chardonnay; (2) Cabernet Sauvignon. (3–4) a 7-day exposure of wine material with isopropanol: (3) Chardonnay; (4) Cabernet Sauvignon. (5) Positive control sample. (6–9) PCR samples of DNA amplified with Vv3-Fwd and Vv3-Rev primers, extracted from precipitated wine debris: (6–7) a 3-day exposure of wine material with isopropanol: (6) Chardonnay; (7) Cabernet Sauvignon. (8–9) a 7-day exposure of wine material with isopropanol: (8) Chardonnay; (9) Cabernet Sauvignon. (10) Positive control sample.

**Figure 4 foods-10-00595-f004:**
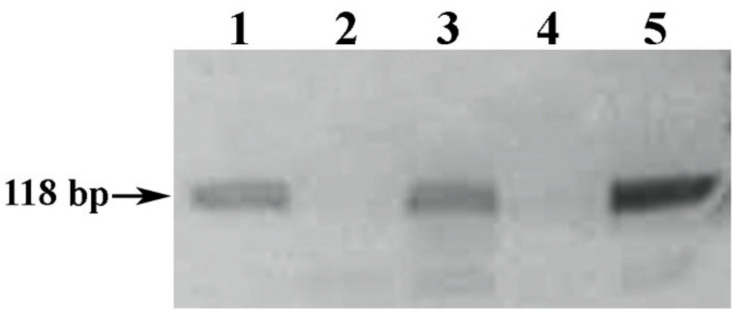
Electrophoregram of the result of PCR amplification of the *Vitis vinifera* L. *UFGT* gene locus with VV3-Fwd and UFGT-R primers. Legend: (1–4) PCR samples of DNA samples amplified with primers Vv3-Fwd and UFGT-R extracted from the deposited wine material debris: (1–2) a 1-h exposure of wine material mixed with isopropanol, sodium acetate and linear polyacrylamide Bioron: (1) Chardonnay; (2) Cabernet Sauvignon. (3–4) 1-h exposure of wine material mixed with isopropanol, sodium acetate and linear polyacrylamide Satellite Red: (3) Chardonnay; (4) Cabernet Sauvignon. (5) Positive control sample.

**Figure 5 foods-10-00595-f005:**
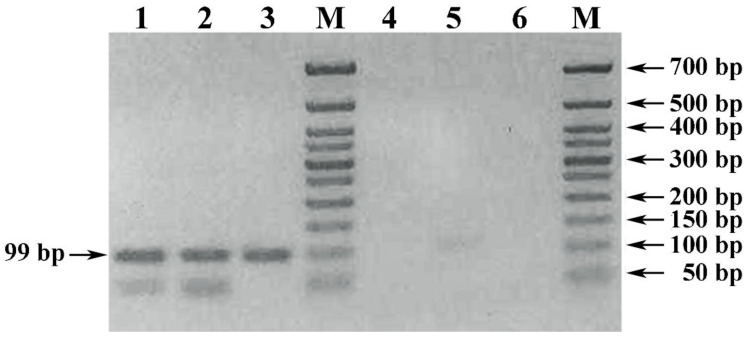
Electrophoregram of results of PCR amplification of the *Vitis vinifera* L. *UFGT* gene locus with UFGT-F1 and UFGT-R1 primers. Legend: (1–3) PCR of a DNA sample extracted by lysis with DMSO of the precipitated debris of young Cabernet Sauvignon wine, prepared by a 10 min exposure of the wine material mixed with PVP and isopropanol. (4–6) PCR of a DNA sample extracted by lysis with DMSO of the precipitated debris of young Cabernet Sauvignon wine, sampled by a 10 min exposure of the wine material mixed with PVP. M) DNA size markers 50+ bp (Evrogen JSC, Moscow, Russia).

**Figure 6 foods-10-00595-f006:**
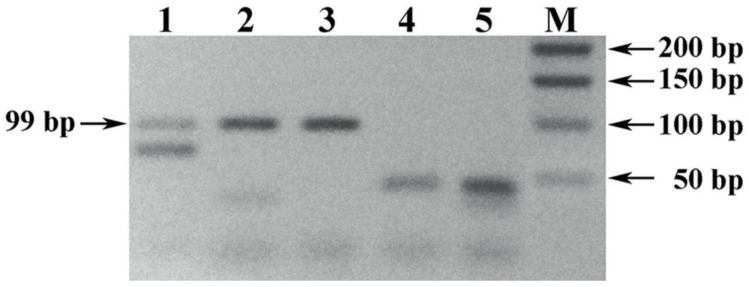
Electrophoregram of the result of PCR amplification of the *Vitis vinifera* L. *UFGT* gene locus with UFGT-F1 and UFGT-R1 primers for reproducibility of the reaction. Legend: (1–2) PCR of a DNA sample extracted with “DNA-sorb-S-M kit”: from the precipitated debris of white wine Visigodo Sauvignon blanc, sampled with a 1-day exposure in a mixture with PVP, sodium acetate and isopropanol. (3–4) PCR of a DNA sample extracted with “DNA-sorb-S-M kit” from the precipitated red wine debris Lagar De Robla (Mencia), sampled by a 10-min exposure in a mixture with PVP and isopropanol. (5) Negative control sample. M) DNA size markers 50+ bp (Evrogen JSC, Moscow, Russia.

**Figure 7 foods-10-00595-f007:**
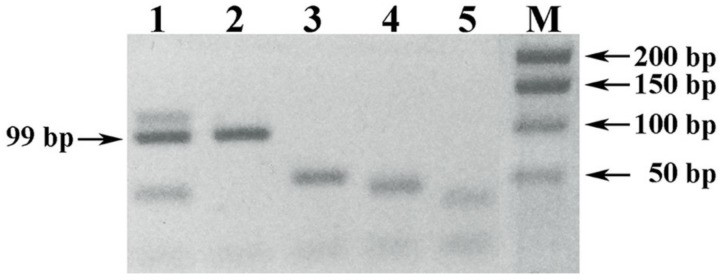
Electrophoregram of the result of PCR amplification of the *Vitis vinifera* L. *UFGT* gene locus with UFGT-F1 and UFGT-R1 primers for the negative effect of cold stabilization on the DNA under research. Legend: (1–2) PCR of a DNA sample extracted with “DNA-sorb-S-M kit” from the precipitated red wine debris Lagar De Robla (Mencia), sampled by a 10-min exposure in a mixture with PVP and isopropanol immediately after uncorking the bottle. (3–4) PCR of a DNA sample extracted with “DNA-sorb-S-M kit” from the precipitated red wine debris Lagar De Robla (Mencia), sampled by a 10-min exposure in a mixture with PVP and isopropanol on the next day after uncorking the bottle and storing it in the refrigerator. (M) DNA size markers 50+ bp (Evrogen JSC, Moscow, Russia).

**Figure 8 foods-10-00595-f008:**
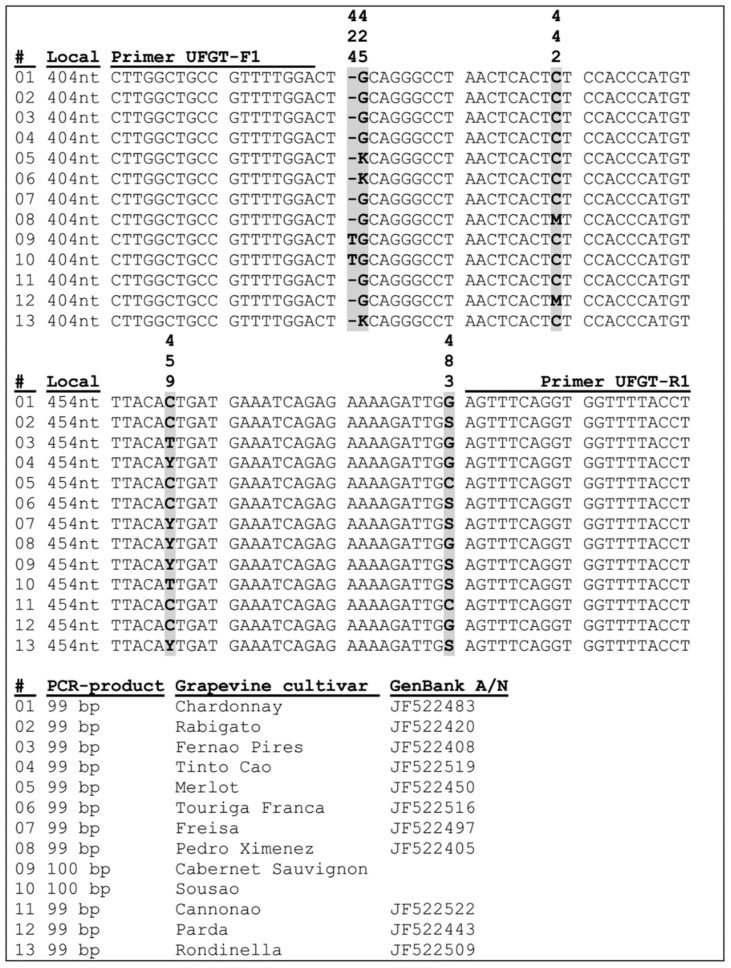
Alignment of reference nucleotide sequences of the *Vitis vinifera* L. *UFGT* gene locus flanked with UFGT-F1 and UFGT-R1 primers. # index number.

**Figure 9 foods-10-00595-f009:**
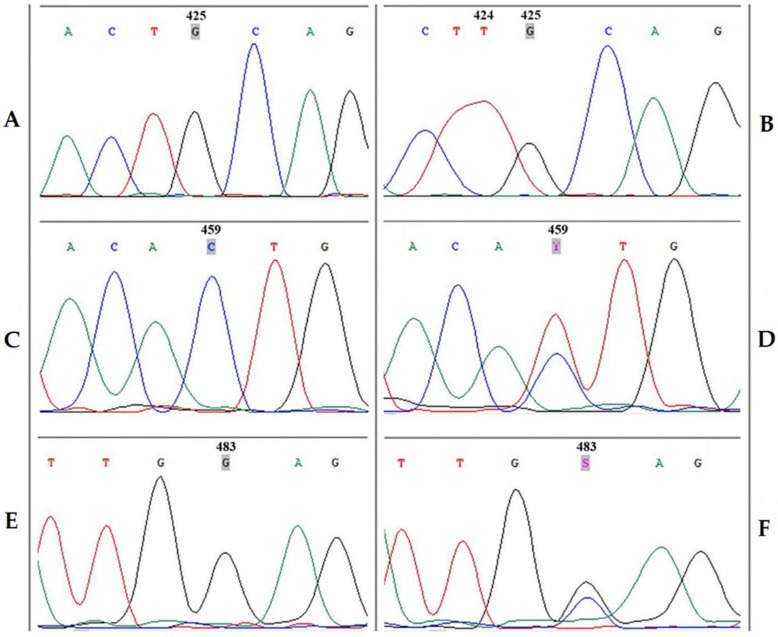
Detection of polymorphic positions (424/425/459/483) of the *Vitis vinifera* L. *UFGT* gene locus by capillary Sanger sequencing method. Legend: (**A**) position 425 (G); (**B**) position 424 (**T**) and 425 (G); (**C**) position 459 (C); (**D**) position 459 (Y); (**E**) position 483 (G); (**F**) position 483 (S).

**Table 1 foods-10-00595-t001:** Elements of sample preparation stages by precipitation of wine debris by centrifugation with pre exposure with precipitators and co-precipitators.

I	II	III	IV	V
15 mL Falcon-type test tube	15 mL Falcon-type test tube	15 mL Falcon-type test tube	15 mL Falcon-type test tube	1.5 mL Eppendorf-type test tube
8 mL wine	7 mL wine	7 mL wine	7 mL white wine	100 μL red wine
-	-	15 μL linear polyacrylamide solution	10 mg PVP	10 mg PVP
-	700 μL 3 M sodium acetate	700 μL 3 M sodium acetate	700 μL 3 M sodium acetate	-
6 mL cold isopropanol	7 mL cold isopropanol	7 mL cold isopropanol	7 mL cold isopropanol	900 μL cold isopropanol
−20 °C 1–14 days	−20 °C 1–3 days	−20 °C 1–24 h	−20 °C 1–24 h	20 °C 10 min, 1–10 days
3000 rpm * 1 h	3000 rpm * 1 h	3000 rpm * 1 h	3000 rpm * 1 h	15000 rpm ** 10 min

* applied to low-speed centrifuge “CM-6M” (Elmi, Riga, Latvia). ** applied to microcentrifuge “CM-50” (Elmi, Riga, Latvia).

**Table 2 foods-10-00595-t002:** Stages of proven methods of DNA extraction from precipitated wine debris.

Extraction with the “DNA-Sorb-S-M Kit”in Modification	Extraction by Dimethyl Sulfoxide Lysis (DMSO)
Resuspension of wine debris in 521 µL of lysing solution (400 µL of buffer for lysing reagent, 17 µL of lysing reagent, 4 µL of 2-mercaptoethanol, 100 µL of proteinase K) by tube vortexing	Resuspending of wine debris in 300 µL 100% DMSO by tube vortexing
	
Incubation of the resultant mixture at 64 °C for 60 min with periodic stirring on the vortex every 10–12 min	Incubation of the resultant mixture at 64 °C for 30 min. with periodic stirring on the vortex every 10 min
	
Precipitation of undissolved sample particles by centrifugation at 10,000× *g* for 5 min	Precipitation of undissolved sample particles by centrifugation at 10,000× *g* for 10 min
	
Selection of the lysate supernatant (400–450 µL) and transfer to a test tube with a sorbent (25 µL)	Selection and transfer of the lysate supernatant (250 µL) into a new test tube
	
Tube vortexing, exposure in a rack for 10 min with periodic stirring on the vortex every 2 min	Mixing of lysate with 250 µL of 4 M ammonium acetate and 1 mL of cold isopropanol (1:1:4 ratio)
	
Centrifugation at 2000× *g* for 1 min	Exposure of the resultant mixture at −20 °C for 60 min
	
Removal of the supernatant	Precipitation of the nucleoprotein complex by centrifugation at 10,000× *g* for 10 min
	
Resuspending of the sorbent in 300 µL of washing solution 1 by vortexing the test tube	Removal of the supernatant
	
Centrifugation at 2000× *g* for 1 min	Insertion of 500 µL of cold 70% ethanol to the deposited nucleoprotein complex
	
Removal of the supernatant	Vortexing the test tube. Exposure in a rack for 10 min with periodic stirring on the vortex every 5 min
	
Resuspending of the sorbent in 500 µL of the solution for washing 2 by vortexing the test tube	Centrifugation at 10,000× *g* for 10 min
	
Centrifugation at 7000× *g* for 1 min	Removal of the supernatant
	
Removal of the supernatant	Incubation of test tubes with open cap at 64 °C for 5–10 min
	
Resuspending of the sorbent in 500 µL of the solution for washing 2 by vortexing the test tube	Resuspending of the dried sediment in 25 µL of TE buffer by incubation at 64 °C for 10 min with periodic stirring on the vortex every 2 min
	
Centrifugation at 7000× *g* for 1 min	Centrifugation at 10,000× *g* for 10 min
	
Incubation of test tubes with open cap at 64 °C for 5–10 min	Transfer of the supernatant to a new test tube
	
Resuspending of the sorbent in 25 µL buffer for elution in a vortexing tube

Incubation of the resultant mixture at 64 °C for 5–10 min with periodic stirring on the vortex every 1 min

Centrifugation at 10,000× *g* for 1 min

Transfer of the supernatant to a new test tube

Transfer of the supernatant to a new test tube

**Table 3 foods-10-00595-t003:** Protocols for preparing reaction mixtures for PCR with DNA samples extracted from grapes and wine debris.

Reagents	PCR with DNA Samples Extracted from Grapes			PCR with DNA Samples Extracted from Wine Debris		
	Initial Concentration	Working Concentration	1 Test (µL)	Initial Concentration	Working Concentration	1 Test (µL)
Sterile water	-	-	16.5	-	-	15
5× Encyclo Red buffer	5×	1×	5	5×	1×	5
50× mixture dNTP	50×	1×	0.5	50×	2×	1
Primer No. 1	50 μM	0.5 μM	0.25	50 μM	1 μM	0.5
Primer No. 2	50 μM	0.5 μM	0.25	50 μM	1 μM	0.5
50× mixture of polymerases Encyclo	50×	1×	0.5	50×	2×	1
DNA matrix	-	-	2	-	-	2 *
Final volume	-	-	25	-	-	25

* the volume of the DNA sample introduced into the reaction polymerase chain reaction (PCR) mixture could vary from 2 to 5 µL.

**Table 4 foods-10-00595-t004:** Oligonucleotide primers and thermocycling modes used in PCR for the genetic identification of grape varieties and DNA authentication of wines produced from them.

Gene Locus	Name and Sequence of Oligonucleotide Primers	Basic Thermal Cycling Mode *	PCR-Product
UFGT	Vv1-Fwd: 5/-GCAATGTAATATCAAGTCC-3/ [30]	×1: 95 °C—300 s; ×40: 95 °C—30 s, 58 °C—30 s, 72 °C—30 s; ×1: 72 °C—300 s	705 bp
	Vv1-Rev: 5/-TTTCTTTCTTTGAGCCATT-3/ [30]		
	Vv3-Fwd: 5/-AGCAGAGATGGGGGTGGCTT-3/ [30]	×1: 95 °C—300 s; ×40: 95 °C—30 s, 58 °C—30 s, 72 °C—30 s; ×1: 72 °C—300 s	119 bp
	Vv3-Rev: 5/-AGCAGGTAAAACCACCTGAA-3/ [30]		
	Vv3-Fwd: 5/-AGCAGAGATGGGGGTGGCTT-3/ [30]	×1: 95 °C—300 s; ×40: 95 °C—10 s, 62 °C—10 s, 72 °C—10 s; ×1: 72 °C—300 s	118 bp
	UFGT-R: 5/-GCAGGTAAAACCACCTGAAACT-3/		
	UFGT-F: 5/-CTTGGCTGCCGTTTTGGACT-3/	×1: 95 °C—300 s; ×40: 95 °C—10 s, 62 °C—10 s, 72 °C—10 s; ×1: 72 °C—300 s	101 bp
	UFGT-R: 5/-GCAGGTAAAACCACCTGAAACT-3/		
	UFGT-F1: 5/-CTTGGCTGCCGTTTTGGA-3/	×1: 95 °C—300 s; ×40: 95 °C—10 s, 58 °C—10 s, 72 °C—10 s; ×1: 72 °C—300 s	99 bp
	UFGT-R1: 5/-AGGTAAAACCACCTGAAACT-3/		
F3H1	F3H_H1fwd: 5/-AGAGAAAGAAGGCGACGT-3/ [31]	×1: 95 °C—300 s; ×40: 95 °C—30 s, 58 °C—30 s, 72 °C—30 s; ×1: 72 °C—300 s	375 bp
	F3H_H1rev: 5/-GATGGCTGGAAACGATGA-3/ [31]		
	F3H_H2fwd: 5/-CTGTTGAAGGAGCTTTCG-3/ [31]	×1: 95 °C—300 s; ×40: 95 °C—30 s, 58 °C—30 s, 72 °C—30 s; ×1: 72 °C—300 s	532 bp
	F3H_H2rev: 5/-GGCTTGGACTCTAACTTG-3/ [31]		

* applied to thermal cycler “Tertsik” (DNA-technology, Russia).

**Table 5 foods-10-00595-t005:** Distribution of technical grape varieties according to the profile of polymorphic positions of the *Vitis vinifera* L. *UFGT* gene locus.

#		Technical Grape Varieties	Polymorphic Positions (INDEL/SNPs)				
			424 *	425	442	459	483
1	B *	Chardonnay	-	G	C	C	G
		Gouveio					
	N *	Touriga Brasileira					
		Donzelinho Tinto					
2	B	Codega do Larinho	-	G	C	C	S
		Rabigato					
	N	Tinta Amarela					
		Alicante Bouschet					
3	B	Fernao Pires	-	G	C	T	G
		Malvasia Fina					
	N	Tinta Roriz					
		Agostana Nera					
4	B	Moscatel Galego	-	G	C	Y	G
		Bianca					
	N	Tinto Cão					
		Pinot Noir					
5	B	Trebbiano RomagnoloVermentino	-	K	C	C	C
	N	Merlot					
		Tinta Barroca					
6	B	Touriga Franca	-	-	-	-	-
	N	Touriga Nacional	-	K	C	C	S
7	B	Viosinho	-	G	C	Y	S
	N	Freisa					
8	B	Pedro Ximenez	-	G	M	Y	G
	N	Rufete					
9	B	-	T	G	C	Y	S
	N	Cabernet Sauvignon					
10	B	-	T	G	C	T	S
	N	Souzão					
11	B	-	-	G	C	C	C
	N	Cannonao					
		Nero d’Avola					
12	B	Parda	-	G	M	C	G
		Blanca Cayetana					
	N	-					
13	B	-	-	K	C	Y	S
	N	Rondinella					
		Raboso Piave					

* B—blanc (white color of grape berry), N—noir (black color of grape berry), INDEL (424 nt). # index number of *UFGT* gene-associated groups.

**Table 6 foods-10-00595-t006:** Distribution of technical grape varieties according to *UFGT* gene-associated groups.

#		GenBank A/N (Technical Grape Varieties, Country)
1	B	JF522533 (Perla, Italy), JF522518 (Malvasia del Chianti, Italy), JF522500 (Robolla, Greece), JF522495 (Matilde, Italy), JF522483 (Chardonnay, Italy), JF522431 (Feteasca Alba, Romania), JF522413 (Sultanina, Greece), JF522412 (Verduzzo Friulano, Italy), JF522411 (Prosecco Balbi, Italy), JF522406 (Ribolla, Italy)
	N	KY293689 (Yaghuty, Iran), KY305474 (Sirch, Iran), KY305473 (Shiraz, Iran), JF522524 (Dolcetto, Italy), JF522512 (Tintoria, Italy), JF522505 Bastardo, Portugal), JF522484 (Franconia, Italy), JF522470 (Carignan, Italy), JF522440 (Colorino, Italy), JF522386 (Monastrel, Spain), JF522380 (Croatina, Italy)
2	B	JF522420 (Rabigato, Portugal)
	N	JF522514 (Tinta Francisca, Portugal), JF522429 (Feteasca Neagra, Romania), JF522417 (Pignola, Italy)
3	B	KJ495698 (Trebbiano Toscano, Italy), KJ495697 (Grechetto, Italy), KJ495695 (Pecorino, Italy), JF522535 (Albarino, Spain), JF522517 (Malvasia Istriana, Italy), JF522501 (Incrocio Manzoni Bianco, Italy), JF522493 (Riesling Italico, Italy), JF522432 (Greco, Italy), JF522426 (Fiano, Italy), JF522401 (Vittoria, Italy), JF522385 (Xarello, Spain), JF522388 (Assyrtiko, Greece), JF522459 (Sauvignon Blanc, Italy), JF522408 (Fernao Pires, Portugal)
	N	KJ495696 (Sangiovese VCR4, Italy), JF522510 (Sangiovese cR10, Italy), JF522506 (Agostana Nera, Italy), JF522486 (Malbech, Italy), JF522472 (Aglianico, Italy), JF522455 (Marzemina Cenerenta, Italy), JF522456 (Marzemina Nera, Italy), JF522454 (Pattaresca, Italy), JF522453 (Refosco dal Peduncolo Rosso, Italy), JF522446 (Ciliegiolo, Italy), JF522436 (Teroldego, Italy), JF522427 (Petit Verdot, Italy), JF522415 (Primitivo di Gioia, Italy), JF522399 (Moscato di Amburgo, Italy), JF522397 (Castelao, Portugal), JF522400 (Mencia, Italy)
4	B	JF522528 (Bianca, Italy), JF522527 (Picolit, Italy), JF522507 (Gatta, Italy), JF522502 (Riesling Renano, Italy), JF522496 (Mustoasa de Maderat, Romania), JF522491 (Pinot Blanc, Italy), JF522475 (Malvasia Fina, Portugal), JF522466 (Tocai, Italy), JF522461 (Albana, Italy), JF522457 (Marzemina Bianca, Italy), JF522438 (Regina, Italy), JF522435 (Grechetto, Italy), JF522424 (Feteasca Regala, Romania), JF522422 (Moscato Bianco, Italy), JF522423 (Moscato Giallo, Italy), JF522410 (Parellada, Spain), JF522402 (Italia, Italy)
	N	JF522519 (Tinto Cão, Portugal), JF522511 (Gruaja, Italy), JF522489 (Pinot Noir, Italy), JF522488 (Marzemina Nera Bastarda, Italy), JF522485 (Xinomavro, Greece), JF522409 (Malvasia Nera, Italy), JF522474 (Alfrocheiro, Portugal), JF522447 (Canaiolo Nero, Italy), JF522468 (Raboso Piave, Italy), JF522469 (Raboso Veronese, Italy), JF522441 (Agiorgitiko, Greece), JF522434 (Corbinona, Italy), JF522433 (Corbinella, Italy), JF522428 (Bovale Sardo, Italy), JF522425 (Lambrusco Maestri, Italy), JF522418 (Moschomavro, Greece), JF522407 (Graciano, Spain), JF522398 (Cardinal, Italy)
5	B	JF522508 (Trebbiano Romagnolo, Italy), JF522499 (Vermentino, Italy)
	N	JF522378 (Tinta Barroca, Portugal), JF522462 (Piedirosso, Italy), JF522450 (Merlot, Italy), JF522391 (Malbo Gentile, Italy)
6	B	-
	N	JF522525 (Corvina, Italy), JF522516 (Touriga Franca, Portugal), JF522448 (Touriga Nacional, Portugal), JF522464 (Carmenere, Italy)
7	B	-
	N	JF522497 (Freisa, Italy)
8	B	JF522405 (Pedro Ximenez, Spain)
	N	-
11 *	B	-
	N	JF522522 (Cannonao, Spain), JF522515 (Tocai Rosso, Italy), JF522442 (Nero d’Avola, Italy)
12	B	JF522443 (Parda, Spain); JF522444 (Blanca Cayetana, Spain); JF522377 (Antao Vaz, Italy)
	N	-
13	B	-
	N	JF522509 (Rondinella, Italy), JF522430 (Cabernet Franc, Italy), JF522476 (Raboso Piave, Italy), JF522477 (Raboso Veronese, Italy), JF522445 (Negrara Veronese, Italy), JF522471 (Nebbiolo, Italy), JF522384 (Cabernet Sauvignon, Italy)

* 9 and 10 has no GenBank database representation. # index number of *UFGT* gene-associated groups.

## Data Availability

Data is contained within the article.

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
