# Peer review of "Methodological Approaches to DNA Authentication of Foods, Wines and Raw Materials for Their Production"

_foods, 2021, doi:10.3390/foods10030595_

Round 1

Reviewer 1 Report

1. Commercial wine were filter sterilized?

2. With the methods applied in this work, is it possible to recover DNA from a wine that has been filtered at 0.2 microns before bottling?

Author Response

  1. Commercial wine were filter sterilized?

    Yes, but most likely with standard filters from 0,45 to 5,0 microns when preparing for bottling.
  2. With the methods applied in this work, is it possible to recover DNA from a wine that has been filtered at 0.2 microns before bottling?

    An additional research is needed for an objective response to the given question. However, it can be assumed that the residual amount of extracted Vitis vinifera L. nucleic acid can be critical for wine authentication, especially if the genetic target for analysis is a unique or low-copy coding sequence of nuclear DNA.

Reviewer 2 Report

The authors used as “Objects of research: technical (wine) grape varieties, experimental young wines, commercial white and red monosort dry wines of protected geographical indications and protected names of places of origin”, applied five sample preparation methods (described in literature or with some modifications) and extracted DNA mostly using a commercial kit. To evaluate the quality of the DNA extracted they used a kit (except primers) to amplify the DNA. The PCR products were visualized after agarose electrophoresis and also sequenced and analysed for detection of INDELs and SNPs.

The authors found that “Sequenced PCR products of DNA samples extracted from the deposited debris of the experimental wine material had a profile of the analyzed polymorphic positions of the UFGT gene locus identical to the raw grape material used in their production. However, the results of sequencing of PCR products of DNA samples extracted from the deposited debris of commercial wines did not always correspond to the variety-authentic profile, which could be caused by external and internal factors.”

In the Discussion the data presented in reference [3] should be part of it.

Lev A. Oganesyants, Ramil R. Vafin, Aram G. Galstyan, Vladislav K. Semipyatniy, Sergey A. Khurshudyan, and Anastasia E. Ryabova. Prospects for DNA authentication in wine production monitoring. Foods and Raw Materials. 2018, vol. 6, no. 2, pp. 438–448. DOI: https://doi.org/10.21603/2308-4057-2018-2-438-448.

The conclusions are not directly related to the main data of the work, they are too general.

  1. Line 28: Consider to remove PCR as a keyword and instead change “DNA extraction” to DNA extraction and amplification as keyword
  2. Line 115 – In Vitis vinifera L change the L to L.
  3. Line 59 – Apply the correct unit for ºC
  4. Table 1 – Change 1,5 ml to 1.5 ml
  5. Table 1 – In 3M introduce a space in order to be 3 M (a space before de unit must exist)
  6. Table 1 – For hour use the unit h
  7. Table 2 – Correct the unit min. to min (3 times incorrect)
  8. Table 2 – In 4M introduce a space in order to be 4 M
  9. Table 3 – Change 16,5 – 0,25 – 0,5 - to 16.5 – 0.25 – 0.5
  10. Table 4 – Correct the unit s. to s (several times incorrect)
  11. Line 223 – Since Table 3 is related to “Protocols for preparing reaction mixtures for PCR with DNA samples extracted from grapes and wine debris” the one to be mentioned here should be Table 4.
  12. Line 235 – Correct UFGT-F and UFGT-R primers to Vv3-Fwd and Vv3-Rev primers
  13. Line 253 – Correct VV3-Fwd to Vv3-Fwd
  14. Line 260 – Correct Table. 1 to Table 1
  15. Lines 276, 279, 282, 294 – Change 10-minute to 10 min since the unite for minutes is min
  16. Line 284 – Correct the unit min. to min
  17. Line 337, 341 – SNP 442 is presented in Table 5 but not in Fig 9
  18. Table 5 – What is the mean of * next to 424* and INDEL*?
  19. Table 5, line 4 – The grape variety is Tinto Cão
  20. Table 5, line 6 – The grape variety is Touriga Nacional
  21. Table 5, line 10 – The grape variety is Souzão
  22. Line 388 – Change 15-ml to 15 ml

Author Response

In the Discussion the data presented in reference [3] should be part of it.

The article (Oganesyants L.A. et al., 2018) was mainly a review article, so we decided to refer to it only in the “Introduction”, while quoting a number of primary sources from that article in the “Discussion”.

The conclusions are not directly related to the main data of the work, they are too general.

“Conclusions” has been improved with the introduction of thematic content directly related to the main data of the research.

The remaining 22 comments on the text of the article are corrected. Please see the revised manuscript, where each correction is numbered using Word's Comment feature.

Reviewer 3 Report

The manuscript entitled “Methodological approaches to DNA authentication of wines and raw material for their production” presents the set up of a reliable and reproducible method for grape variety identification in wines and raw material.

Nowadays, several methods for the extraction of good quality DNA from wine and raw material are available, so I think that the first part of the results is pointless. However, I found interesting the choice of UFGT gene for grape variety traceability. The availability of different alleles makes this locus useful for the aim of the study. However, the Sanger sequencing is a slow and high-cost method; the detection of allele could be easily performed with other methods, such as CAPS or allele-specific primer design. I suggest focusing on this aspect of the study.

Author Response

Considering the potential and limitations of existing methods of extraction of residual DNA of Vitis vinifera L. from wine, the research on improving this procedure is still highly relevant. The current level of techniques of the well-known methods for extraction of residual grapevine nucleic acids from commercial wines has not yet led to the creation of a specialized commercial kit capable of providing standardized DNA extraction and subsequent reproducibility of genetic tests. We plan to use other molecular genetic methods (including CAPS and AS-PCR) in further research.

Round 2

Reviewer 3 Report

Currently, several efficient methods for DNA extraction from commercial wines are available: Pereira et al., 2011 (DOI: 10.5344/ajev.2011.10022), Boccacci et al., 2012 (DOI: 10.1007/s00217-012-1770-3), Bigliazzi et al., 2012 (DOI: 10.5344/ajev.2012.12014). In my opinion, the work does not contribute significantly to existing knowledge in the field.

Author Response

The above-mentioned literature mainly contains CTAB-modified methods for extracting nucleic acids of Vitis vinifera L. from wines, a number of which (Pereira et al., 2011; Boccacci et al., 2012) require a long sample preparation procedure by exposing a mixture of wine and precipitator (isopropanol) in a freezer for two weeks. Pereira et al. (2011) explain the reason for such a long exposure period as follows: “all white wine samples were amplifiable by PCR, whereas red wine samples were only amplifiable when precipitated for two weeks. Thus, precipitations for two weeks were preferred to all other time-scales” (DOI: 10.5344/ajev.2011.10022).

The TECP-method (Bigliazzi et al., 2012) includes additional manipulations using the commercial QIAprep Spin Miniprep Kit, which significantly increases the number of stages of DNA extraction.

At the same time, Catalano V. et al. (2016) have tested DNA extraction methods for efficiency, including a number of the above-mentioned methods, and indicated that “All the classical DNA extraction methods discussed above are long and laborious” (DOI: 10.1021/acs.jafc.6b02560).

Besides, the methods of nucleic acid extraction of Vitis vinifera L. from wines tested in their work (Catalano V. et al., 2016) still did not provide the proper reproducibility of PCR for the successful interpretation of the results of the analysis, which also indicates the need for further development of the key elements of DNA authentication (sample preparation, DNA extraction, PCR amplification and other molecular genetic research methods).